# Protamine-2 Deficiency Initiates a Reactive Oxygen Species (ROS)-Mediated Destruction Cascade during Epididymal Sperm Maturation in Mice

**DOI:** 10.3390/cells9081789

**Published:** 2020-07-27

**Authors:** Simon Schneider, Farhad Shakeri, Christian Trötschel, Lena Arévalo, Alexander Kruse, Andreas Buness, Ansgar Poetsch, Klaus Steger, Hubert Schorle

**Affiliations:** 1Department of Developmental Pathology, Institute of Pathology, University Hospital Bonn, 53127 Bonn, Germany; simon.schneider@ukbonn.de (S.S.); lena.arevalo@ukbonn.de (L.A.); 2Institute for Medical Biometry, Informatics and Epidemiology, Medical Faculty, University of Bonn, 53127 Bonn, Germany; farhad.shakeri@uni-bonn.de (F.S.); andreas.buness@uni-bonn.de (A.B.); 3Institute for Genomic Statistics and Bioinformatics, Medical Faculty, University of Bonn, 53127 Bonn, Germany; 4Department of Plant Biochemistry, Ruhr-University Bochum, 44801 Bochum, Germany; christian.troetschel@rub.de (C.T.); ansgar.poetsch@rub.de (A.P.); 5Department of Urology, Pediatric Urology and Andrology, Section Molecular Andrology, Biomedical Research Center of the Justus-Liebig University Gießen, 35392 Gießen, Germany; Alexander.Kruse@chiru.med.uni-giessen.de (A.K.); klaus.steger@chiru.med.uni-giessen.de (K.S.); 6Laboratory for Marine Biology and Biotechnology, Qingdao National Laboratory for Marine Science and Technology, Qingdao 266237, China; 7College of Marine Life Sciences, Ocean University of China, Qingdao 266003, China

**Keywords:** protamines, sperm, infertility, ROS

## Abstract

Protamines are the safeguards of the paternal sperm genome. They replace most of the histones during spermiogenesis, resulting in DNA hypercondensation, thereby protecting its genome from environmental noxa. Impaired protamination has been linked to male infertility in mice and humans in many studies. Apart from impaired DNA integrity, protamine-deficient human and murine sperm show multiple secondary effects, including decreased motility and aberrant head morphology. In this study, we use a Protamine*-2* (*Prm2*)-deficient mouse model in combination with label-free quantitative proteomics to decipher the underlying molecular processes of these effects. We show that loss of the sperm’s antioxidant capacity, indicated by downregulation of key proteins like Superoxide dismutase type 1 (SOD1) and Peroxiredoxin 5 (PRDX5), ultimately initiates an oxidative stress-mediated destruction cascade during epididymal sperm maturation. This is confirmed by an increased level of 8-OHdG in epididymal sperm, a biomarker for oxidative stress-mediated DNA damage. *Prm2*-deficient testicular sperm are not affected and initiate the proper development of blastocyst stage preimplantation embryos in vitro upon intracytoplasmic sperm injection (ICSI) into oocytes. Our results provide new insight into the role of *Prm2* and its downstream molecular effects on sperm function and present an important contribution to the investigation of new treatment regimens for infertile men with impaired protamination.

## 1. Introduction

Male fertility relies on the tightly regulated differentiation of spermatogonial stem cells into flagellated sperm within the seminiferous epithelium of the testis. During spermatogenesis, a series of mitotic and meiotic divisions results in the formation of haploid, round spermatids. Extensive cytological remodeling during spermiogenesis generates the characteristic, species-specific sperm morphology. Canonical histones are replaced by testis-specific histone variants and transition proteins, which finally guide the deposition of protamines into the DNA [1]. This induces a conformational change from a nucleosomal into a toroidal chromatin structure, with few remaining histone solenoid structures, thereby conferring DNA hypercondensation [2,3]. Chromatin remodeling is pivotal for sperm function as it is supposed to protect the paternal genome from environmental noxa, to induce a transcriptionally quiescent state and to contribute to the hydrodynamic shape of the sperm head [4].

Protamines are characterized by a high content of arginine- and cysteine-rich domains. While positively charged arginine residues mediate binding to the negatively charged DNA, alternating cysteine residues mediate the formation of intra- and intermolecular disulfide bridges, thereby strengthening DNA compaction [5,6]. In humans and many rodents two protamines, Protamine-1 (Prm1) and Protamine-2 (Prm2) are expressed in species-specific ratios [7]. Aberrations of the ratio have been correlated with subfertility in men in many studies [8,9,10,11,12,13,14,15,16]. Apart from defects in DNA hypercondensation, impaired protamination was additionally correlated with increased DNA damage, reduced sperm motility, decreased viability and acrosomal malformations [17,18,19,20,21,22]. In consequence, infertile men with irregular protamination show decreased fertilization rates in ART (Assisted Reproductive Technologies) programs [23,24,25].

Since cell culture models for human spermatogenesis are scarce and inefficient, mouse models have contributed to a broader understanding of protamine function. Several knockout studies for both, *Prm1* as well as *Prm2,* have shown that both isoforms are indispensable for male fertility in mice as well [26,27,28,29]. Similar to human sperm, impaired protamination was linked to faulty DNA compaction and increased DNA fragmentation. Further, sperm displayed secondary defects like a decline in motility and morphological abnormalities.

In this study, we used a *Prm2*-deficient mouse model established by us to study the molecular origin of these defects using sperm proteomics.

## 2. Materials and Methods

### 2.1. Animals

All animal experiments were conducted according to German law of animal protection and in agreement with the approval of the local institutional animal care committees (Landesamt für Natur, Umwelt und Verbraucherschutz, North Rhine-Westphalia, Germany, approval ID: AZ84-02.04.2013.A429, approved: 13.02.2014). *Prm2*-deficient mice (MGI: 5760133; 5770554) were generated and characterized as published previously [29].

### 2.2. Protein Isolation from Tissue

Fresh tissue samples were taken up in RIPA buffer (Cell Signaling Technology, Cambridge, UK) supplemented with protease inhibitor cocktail (cOmplete™ ULTRA, Roche, Basel, Switzerland) and minced in a glass dounce homogenizer. Samples were sonicated (Bioruptor, Diagenode, Seraing, Belgium) for 5 min at 30 s intervals after incubation on ice for 15 min. Cell debris was removed by centrifugation (20,800× *g*, 30 min, 4 °C). Protein concentrations were quantified using the Pierce™ BCA Protein Assay Kit (Thermo Fisher Scientific, Waltham, MA, USA).

### 2.3. Protein Isolation from Sperm

Isolated sperm were washed once with PBS containing protease inhibitors and pelleted by centrifugation (800 g, 7 min, RT). Next, sperm were dissolved in H_2_O supplemented with protease inhibitors to induce lysis of contaminating non-sperm cells (e.g., blood cells) by hypotonic shock. Sperm pellets were dissolved in RIPA buffer (Cell Signaling Technology, 10 µL per 1 × 10^6^ sperm) supplemented with protease inhibitor cocktail and 100 mM DTT. Following incubation on ice for 30 min, samples were sonified for 5 min in 30 s intervals using a Bioruptor sonication device (Diagenode, Seraing, Belgium). Samples were diluted in Roti^®^-Load 1 and boiled at 95°C for 5 min. Cell debris was removed by centrifugation (20,800× *g*, 4 °C). Protein concentrations were quantified using the Pierce 660 nm Protein Assay Reagent supplemented with Ionic Detergent Compatibility Reagent according to the 96-well microplate procedure of the manufacturer’s protocol assay (Thermo Fisher Scientific).

### 2.4. Immunoblotting

Protein lysates were separated on 12% SDS polyacrylamide gels and transferred to PVDF membranes using the Trans-Blot Turbo System (BioRad, Feldkirchen, Germany). Equal protein loading and successful blotting was verified by Coomassie Brilliant Blue staining. Membranes were blocked in 5% non-fat dry milk powder in phosphate buffered saline with 1% Tween20 (PBS-T) for 1.5 h at room temperature. Primary and secondary antibodies were diluted in blocking solution as specified in Table 1 and incubated at 4°C overnight and 1 h at room temperature, respectively. Following incubation with Pierce Super Signal West Pico chemiluminescent substrate (Perbio, Bonn, Germany), chemiluminescent signals were detected with ChemiDocMP Imaging system (BioRad).

### 2.5. Mass Spectrometry

Following separation of protein extracts by SDS-PAGE, protein gels were shortly washed with _dd_H_2_O and stained with colloidal Coomassie overnight at RT. Gels were washed twice with 1% (*v*/*v*) acetate in _dd_H_2_O and protein lanes were cut into 12 fractions according to their molecular weight. Each fraction was further chopped into squares of around 1 mm^2^, transferred to protein low-binding tubes and fixed with 150 µl 50% (*v*/*v*) MeOH. The fixation solution was replaced by 150 µL destaining solution, followed by incubation for 20 min at 37°C under vigorous shaking. The destaining step was repeated 2–3× until gel pieces appeared clear and gel slices were dried in a vacuum centrifuge. In-gel digestion of proteins was performed with trypsin solution followed by separation of peptides by liquid chromatography (LC) in a gradient using HSS-T3 analytical columns (1.8 µm particle, 75 µm × 150 mm) in combination with the nanoAcquity LC pump system. Peptides were analyzed on a LTQ Orbitrap Elite mass spectrometer (Thermo Fisher) as described [30] and raw data were processed with Proteome Discoverer software version 2.3.0.523 (Thermo Fisher).

Peptide identification was done with an in-house Mascot server version 2.6.1 (Matrix Science Ltd., London, UK). MS data were searched against the mouse reference proteome from Uniprot (2019/05) and a contaminants database (cRAP) [31]. Precursor ion m/z tolerance was 10 ppm, fragment ion tolerance 0.5 Da. Tryptic peptides with up to two missed cleavage were searched. Propionamidylation of cysteines was set as static modification. Oxidation (Met) and acetylation (protein N-termini) were allowed as dynamic modification. Mascot results were evaluated by the Percolator algorithm [32] version 3.02.1 as implemented in Proteome Discoverer. Spectra with identifications above q-value 0.01 were sent to a second round of database search with semi-tryptic enzyme specificity (one missed cleavage allowed). Propionamide and carbamidomethyl were set as additional dynamic cysteine modification. 362,449 peptide spectrum matches (PSMs) were identified with high confidence out of 1,789,702 spectra. 26,084 peptide groups of 7,432 proteins were included. These were assigned to 3,046 distinguishable protein groups. 2,591,992 features were considered for quantification. Actual FDR values of the whole dataset were 0.4%, 1.0%, and 7.3% for PSMs, peptides, and proteins, respectively.

The mass spectrometry proteomics data have been deposited to the ProteomeXchange Consortium via the PRIDE [33] partner repository with the dataset identifier PXD018843 and 10.6019/PXD018843.

Protein quantification and statistical evaluation were performed in R environment (version 3.5.2 [34] on the PSM level data, exported from Protein Discoverer. First, low quality data, including ambiguous peptides matching to more than one protein and proteins detected by a single peptide only were removed from the dataset. Next, the data were variance-stabilized and transformed using Variance Stabilizing Normalization (VSN) [35]. Protein abundances were calculated for each replicate by summarizing measurements of all peptides linked to the respective protein using Tukey’s median polish procedure. To decipher the overall structure within the dataset, PCA analysis was performed using the R-package FactoMineR [36]. For statistical analyses, comparative analyses between different sample groups were performed using moderated t-test from the R-package Limma (version 3.40.2) [37]. The resulting p-values were adjusted for multiple testing using the Benjamini-Hochberg method. Proteins with a false discovery rate (FDR) smaller than 0.05 and an absolute fold-change greater than 1 were considered to be significantly differentially regulated. Plots and heatmaps were generated utilizing the published R-packages ggplot2 and Complex heatmaps [38,39].

### 2.6. Immunohistochemistry

Dissected tissues were fixed in Bouin’s solution at 4°C overnight and processed in paraffin wax. Sections of 3 µm thickness were obtained on glass slides, deparaffinized, rehydrated and treated with decondensing buffer (25 mM DTT, 0.2% (*v*/*v*) Triton x-100 and 200 i.U./mL heparin in PBS) for 1 min at 37°C to increase antigen accessibility. Sections were further processed in the Lab Vision PT module (Thermo Fisher Scientific) and Autostainer 480S (Medac, Hamburg, Germany) as published previously [40]. The primary antibody against 8-OHdG (clone 15A3, Santa Cruz Biotechnology, Dallas, TX, USA) was used at a dilution of 1:200.

### 2.7. Enzyme-Linked Immunosorbent Assay (ELISA)

8-OHdG levels were quantified in DNA of epididymal sperm, which was isolated by phenol/chloroform extraction as published elsewhere [41]. ELISA was performed according to the manufacturer`s instructions of the Oxiselect^TM^ Oxidative DNA Damage ELISA Kit (STA-320, Cell Biolabs Inc., Cologne, Germany), including pretreatment of DNA with Nuclease P1 (N8630, Sigma-Aldrich/Merck, Darmstadt, Germany) and alkaline phosphatase (Sigma-Aldrich/Merck). 3 µg of DNA were used as input. The primary antibody (8-OHdG) was used at a dilution of 1:500 and the secondary antibody at a dilution of 1:1000. Absorbance was recorded at 450 nm (Multiskan GO 1510, Thermo Fisher) and interpreted with Curve Expert 1.4 (Hyams Development, www.curveexpert.net). Statistical analysis was performed with GraphPad Prism 8 (GraphPad Software, San Diego, CA, USA).

### 2.8. Sperm Vitality Assessment

Sperm vitality was analyzed by eosin-nigrosin staining and hypoosmotic swelling test as published elsewhere [42].

### 2.9. Periodic Acid Schiff Staining (PAS)

Bouin’s fixed testicular tissue sections were deparaffinized, re-hydrated and incubated with periodic acid (0.5% in H_2_O) for 10 min. Slides were rinsed with H_2_O and treated with Schiff reagent for 20 min. Finally, slides were washed with H_2_O for 7 min, counterstained with Haemalaun, dehydrated and mounted with a coverslip.

### 2.10. Sperm Nuclear Morphology Analysis

For testicular sperm, 5 µm thick testicular cross sections were stained with Mayer’s Haemalum solution (Merck) and mounted with Dako Faramount Aqueous Mounting Medium (Dako, Carpinteria, CA, USA). Digital photographs of elongating and elongated spermatids were taken at 400x magnification. Spermatid length and width were determined using a macro written for ImageJ, which can be made available upon request.

Epididymal sperm were analyzed using the ImageJ plugin “Nuclear morphology analysis v1.14.1” according to the developers instructions [43]. Briefly, isolated sperm were pelleted by centrifugation (500 g, 5 min, RT) and fixed by dropwise addition of methanol:acetic acid (3:1, (*v*/*v*)). Samples were washed three times with fixative, spread on glass slides, air-dried and mounted with ROTI^®^Mount FluorCare DAPI (Carl Roth, Karlsruhe, Germany). Sperm heads were imaged under oil immersion at 100-fold magnification using an epifluorescence microscope (DM5500B, Leica, Wetzlar, Germany) equipped with a 3CCD-camera (KY-F75U, JVC, Yokohama, Japan). At least 140 sperm heads from three biological replicates each were included into the analysis.

### 2.11. RNAseq Analysis

Total RNA was extracted from testicular tissue after removal of the *tunica albuginea* using TRIzol™ reagent according to the manufacturer’s protocol (Thermo Fisher Scientific). RNA integrity (RIN) was determined using the RNA Nano 6000 Assay Kit with the Agilent Bioanalyzer 2100 system (Agilent Technologies, Santa Clara, CA, USA). RIN values ranged from 8.2–10 for all samples. RNA sample quality control and library preparation were performed by the University of Bonn Core facility for Next Generation Sequencing (NGS), using the QuantSeq 3´-mRNA Library Prep (Lexogen, Greenland, NH, USA). RNAseq was performed on the Illumina HiSeq 2500 V4 platform, producing >10 million, 50 bp 3′-end reads per sample.

All reads were mapped to the mouse genome (GRCm38.89) using HISAT2 2.1 [44]. Transcripts were quantified and annotated using StringTie 1.3.3 [45]. Gene annotation information for the mouse genome was retrieved from the Ensembl FTP server (ftp://ftp.ensembl.org) (GRCm38.89). The python script (preDE.py) included in the StringTie package was used to prepare gene-level count matrices for analysis of differential gene expression (DE).

DE was tested with DESeq2 1.16.1 [46]. Pseudogenes were removed from the count matrices based on “biotype” annotation information extracted from Biomart (R-package biomaRt, [47]). Low counts were removed by the independent filtering process, implemented in DESeq2 [48]. The adjusted p-value (Benjamini-Hochberg method) cutoff for DE was set at <0.05, log_2_ fold change of expression (LFC) cutoff was set at >0.5.

We performed GO term and pathway overrepresentation analyses on relevant lists of genes from DE and co-expression analyses using the PANTHER gene list analysis tool with Fisher’s exact test and FDR correction [49]. We tested for overrepresentation based on the GO annotation database (Biological Processes) (released 2 February 2019, [50]) and the Reactome pathway database (version 58 [51]). The RNAseq data discussed in this publication have been deposited in NCBI’s Gene Expression Omnibus [52] and are accessible through GEO Series accession number GSE149454.

### 2.12. Intracytoplasmatic Sperm Injection (ICSI)

B6D2F1 females were superovulated by intraperitoneal injection of 5 i.U. Pregnant Mare Serum (PMS) and human Chorionic Gonadotropin (hCG). Oocytes were isolated from the oviducts 15 h after the last hormone injection and freed from cumulus cells by treatment with hyaluronidase. Testicular sperm were isolated from 8–13 week-old males as described [53]. Briefly, after removal of the tunica albuginea, testes were placed into 1% (*m*/*v*) PVP solution (P5288, Sigma) and cut into minute pieces. One part of the testicular suspension was throughout mixed with two parts 12% PVP solution and incubated at 16 °C until injection.

ICSI was performed at 17 °C using an inverted microscope (Leica, Wetzlar, Germany) equipped with micromanipulators (Narishige, Tokyo, Japan) and a piezo element (Eppendorf, Hamburg, Germany). Injection capillaries (PIEZO 8-15-NS, Origio, Charlottesville, VA, USA) were filled with Fluorinert (FC-770, Sigma) for proper Piezo pulse propagation. Testicular sperm were collected and injected into oocytes as described in detail by Yoshida et al. [54] with minor modifications. Here, testicular sperm were injected head to tail without prior removal of the sperm flagellum. Oocytes were cultured in M2 medium (Sigma) for injection. Surviving oocytes were cultivated in a drop-culture of KSOM (Gynemed, Lensahn, Germany) under mineral oil (Gynemed) at 37 °C and 5% CO_2_.

### 2.13. Genotyping of Blastocysts

DNA was extracted from blastocysts as published [55]. Briefly, single embryos were supplemented with 10 µL embryo lysis buffer (50 mM KCl, 10 mM Tris-HCl pH 8.3, 2.5 mM MgCl_2_, 0.1 mg/mL gelatin, 0.45% (*v*/*v*) NP40, 0.45% (*v*/*v*) Tween-20, 0.1 mg/mL proteinase K) and incubated at 56 °C for 30 min, followed by 95°C for 10 min. Lysates were cooled down to RT and vortexed vigorously. *Prm2* genotyping was conducted as published previously [29] using 5 µL of the lysate as input.

## 3. Results

### 3.1. To the Molecular Origin of Secondary Defects in Prm2-Deficient Sperm

Previous attempts to establish *Prm2*-deficient mouse lines using classical gene-targeting failed as male chimeras generated by blastocyst injection of *Prm2*^+/−^ ES-cells were sterile [26]. This prohibited detailed functional studies on protamine function in mice. Recently, we reported the successful generation and establishment of *Prm2*-deficient mouse lines using CRISPR/Cas9-mediated gene-editing in oocytes [29]. Interestingly, *Prm2*^+/−^ males displayed normal fecundity with sperm being morphologically and functionally indistinguishable from wildtype sperm. This enabled the breeding of *Prm2*^−/−^ males, which are infertile. Comprehensive phenotypical characterization revealed that the loss of *Prm2* did neither affect the efficiency of spermatogenesis nor epididymal sperm counts but resulted in defective DNA integrity in more than 90% of sperm [29]. Furthermore, 100% of sperm were immotile, displayed morphological malformations, membrane defects and a decline in viability [29] (Appendix A). We categorized the latter as secondary defects, as they cannot be explained by the molecular function of protamines. Since incorporation of protamines into the sperm DNA is supposed to induce transcriptional silencing, we expected to find the underlying reason for induced secondary sperm defects rather on proteomic than on transcriptomic level. Therefore, epididymal sperm protein lysates from wildtype, *Prm2*^+/−^ and *Prm2*^−/−^ males were pre-fractionated by SDS gel electrophoresis and subjected to LC-MS (liquid chromatography—mass spectrometry). In total, 3.318 different proteins were identified with high confidence. Following removal of non-unique peptides and single-shot proteins, 2.299 proteins were considered for label-free quantification (Appendix A). In order to evaluate the power of the generated dataset, we compared the list of identified proteins with the compiled murine sperm proteome of four previous studies [56,57,58,59] showing a coverage of 63% (Figure 1A).

Our analysis detected 1,498 additional proteins, which had not been identified previously. Further, we also compared our dataset with the human sperm proteome, which has been studied in much more detail [60]. Of note, 77% of the murine sperm proteome were overlapping with the human sperm proteome (Figure 1A). Thus, we concluded that the present dataset reflects a significant portion of the murine sperm proteome. Following data normalization and quality controls (Appendix A), a principle component analysis was performed. Knockout replicates clustered strongly apart from controls, being indicative for an overall biological difference (Figure 1B). This was verified by heatmap visualization of the most significantly deregulated proteins. Knockout replicates displayed a strongly altered proteomic profile and were only distantly related to control replicates as visualized by hierarchical clustering using the Euclidian distance (Figure 1C). In accordance with the phenotypical characterization, a highly similar proteomic profile was observed for wildtype and *Prm2*^+/−^ replicates.

No proteins were found to be significantly up- or downregulated (Figure 2A). However, in the knockout condition, 24 proteins were found to be significantly deregulated compared to wildtype (17 down- and 8 upregulated) (Figure 2A). A highly similar set of deregulated proteins was observed by pairwise comparison of knockout and heterozygous samples (Figure 2A). Many of the differentially expressed proteins, including SORD, SOD1, PRDX5 and ACR have a well-known function for male fertility and sperm motility [61,62,63,64]. Thus, deregulated proteins correlate with the observed defects. To investigate whether deregulations can be attributed to specific biological processes, reactome pathway analysis was performed, showing a significant enrichment of downregulated proteins in processes of energy metabolism and the detoxification of ROS (Figure 2B). Further, STRING analysis suggests a strong interaction among the downregulated proteins (Figure 2C). Taken together, MS data imply that loss of PRM2 triggers a molecular cascade that initiates described secondary sperm defects, which ultimately result in male infertility.

### 3.2. Prm2 Deficiency Induces a Downregulation of ROS Scavenger Proteins

Since excessive levels of reactive oxygen species (ROS) are well-known mediators of cellular damage, including DNA damage and lipid peroxidation, the potential role of imbalanced ROS level was investigated in detail. Among the differentially expressed proteins, superoxide dismutase 1 (SOD1) and peroxiredoxin 5 (PRDX5) were identified as key players of ROS detoxification. SOD1 catalyzes the conversion of superoxide radicals into hydrogen peroxide, which is subsequently degraded into non-toxic H_2_O molecules. PRDX5 is another cytoprotective antioxidant enzyme that catalyzes the reduction of hydrogen peroxide, alkyl hydroperoxide and peroxynitrite [65]. Thus, both enzymes have ROS scavenging function that might result in oxidative stress upon loss. As shown in the profile plots for SOD1 and PRDX5, the relative abundance of SOD1 and PRDX5 peptides identified by MS was strongly diminished in *Prm2*^−/−^ sperm compared to controls (Figure 3A and Appendix A).

To validate MS results independently, protein level of SOD1 and PRDX5 were determined by Western Blot analysis. Both, levels of SOD1 and PRDX5 were strongly decreased in *Prm2*^−/−^ sperm compared to wildtype and *Prm2*^+/−^ samples (Figure 3B). However, in testicular lysates protein levels were not changed (Figure 3B). This suggests that the proposed ROS signaling cascade is first initiated during epididymal transit.

### 3.3. Oxidative Stress Causes DNA Damage During Epididymal Sperm Maturation

To prove the hypothesis that excessive ROS levels are the cause of secondary sperm defects, testicular and epididymal tissue sections were stained against 8-OHdG (8-hydroxy-2’-deoxyguanosine, 8-hydroxyguanine and 8-hydroxyguanosine). Of note, mature step 16 spermatids of all genotypes stained negative for 8-OHdG, indicative for intact and undamaged DNA (Figure 4A).

However, in *Prm2*^−/−^ males, increasing levels of 8-OHdG positive sperm nuclei were identified during epididymal transit. After release from the testis, sperm first pass through the corpus and caput epididymis. At this stage, 8-OHdG positive nuclei were first detected. Some nuclei were completely stained while others showed a more granular staining (Figure 4A). Still, some sperm nuclei stained negative for 8-OHdG (Figure 4A, arrowheads). With proceeding sperm maturation signals intensified. In tissue sections of the cauda epididymis, almost all sperm nuclei showed a strong staining for 8-OHdG, being highly suggestive for oxidative stress-mediated DNA damage. In contrast, sections of caput and cauda epididymis from *Prm2*^+/+^ and *Prm2*^+/−^ displayed only few 8-OHdG positive cells (Figure 4A, arrows). To quantify the increase in oxidative DNA damage in *Prm2*^−/−^ epididymal sperm, an enzyme-linked immunosorbent assay (ELISA) was performed. The concentration of 8-OHdG was highly significantly increased from 1.449 ng/mL in DNA from wildtype sperm to 3.256 ng/mL in *Prm2*^−/−^ sperm (Figure 4B). Only a slight, but non-significant increase in 8-OHdG concentrations was observed between wildtype and *Prm2*^+/−^ sperm. Taken together, the results clearly show that *Prm2* deficiency leads to an imbalance in the ROS scavenging system, supposedly leading to elevated ROS levels triggering a reactive oxygen-mediated destruction cascade in sperm, which is exerted during epididymal sperm maturation.

### 3.4. Secondary Sperm Defects Arise During Epididymal Maturation

To verify that other secondary sperm defects can be also assigned to the proposed ROS pathway, these defects are expected to appear and intensify during epididymal sperm maturation as well. The initial phenotypical characterization showed severe acrosomal malformations in *Prm2*-deficient cauda epididymal sperm [29]. Here, periodic-acid-Schiff (PAS) staining, which stains glycoprotein-rich acrosomal structures pink, was performed on testicular tissue sections. Acrosomal vesicles were first detected in round spermatids (Figure 5A, arrowheads).

In the course of spermiogenesis, vesicles elongated and finally capped the sickle-shaped head of step 16 spermatids of all genotypes (Figure 5A, arrows). This indicates that acrosome biogenesis is not impaired upon loss of *Prm2* and that defects are inevitably acquired during epididymal transit.

Further, we analyzed the effects of *Prm2* deficiency on the nuclear morphology of spermatids and mature sperm. The characteristic sickle-shaped nuclear morphology was observed in hematoxylin stained stage VII/VIII seminiferous tubules of all genotypes (Figure 5B). Quantification of nuclear head length and width revealed no significant differences between *Prm2*-deficient step 16 testicular spermatids and controls (Figure 5B), thus indicating that *Prm2* is dispensable for proper shaping of the sperm head during spermiogenesis. However, severe changes in sperm nuclear morphology were observed in DAPI stained cauda epididymal sperm (Figure 5C). In general, most *Prm2*-deficient sperm nuclei appeared smaller in size and displayed loss of the characteristic sickle-shape compared to controls (Figure 5C).

However, morphological changes were variable, resulting in a highly heterogeneous sperm population (Figure 5C). To quantify changes in sperm nuclear head morphology, an automated, high-throughput analysis was performed. A consensus shape was computed for each genotype, visualizing the observed morphological changes (Figure 5D). With a median of 10.53 μm^2^ (95% confidence interval (CI) 10.96 μm^2^ ± 0.26), the head size of *Prm2*-deficient sperm was more than 40% reduced compared to controls with a median of 18.42 μm^2^ (95% CI 18.38 μm^2^ ± 0.1) (Appendix A). In line, the median sperm perimeter decreased from 21.13 μm (95% CI 21.09 μm ± 0.08) to 14.21 μm (95% CI 14.76 μm ± 0.25) (Appendix A). Both, a decline of sperm length and width resulted in a decrease of sperm area and perimeter. Further, loss of the characteristic sperm hook caused an increase in circularity and ellipticity of *Prm2*-deficient sperm compared to controls (Appendix A).

Taken together, our analyses clearly demonstrate that the functional and morphological sperm defects observed are almost exclusively acquired during epididymal sperm maturation and correlate with the initiation of the postulated ROS cascade.

To assess whether *Prm2* deficiency affects global testicular gene expression, RNAseq analysis was performed on testicular RNA isolated from *Prm2*^+/+^, *Prm2*^+/−^ and *Prm2*^−/−^ males. Heatmap visualization of the top 50 differentially expressed (DE) genes and hierarchical clustering using the Euclidian distance clearly separated samples of *Prm2*^−/−^ males from controls as also verified by PCA analysis (Figure 6A,B). Most DE genes were stronger expressed in *Prm2*^−/−^ testes compared to controls. Mapping of upregulated genes to their chromosomal localization, showed a random distribution throughout the whole genome and no enrichment at distinct loci (Appendix A). This indicated that the protamine-mediated gene silencing during spermiogenesis might be incomplete upon loss of *Prm2*. In total, 81 genes were found to be upregulated and 13 genes to be downregulated in knockout testes compared to wildtype (Figure 6C). Among the upregulated genes, well-known regulators of protamine expression and translational activation were identified, e.g., Y-box binding protein 2 (*Ybx2*) and TAR binding-protein 2 (*Tarbp2*), probably resembling a compensation mechanism due to the lack of *Prm2*. However, gene ontology analysis on up- and downregulated gene sets did not identify enrichment of any biological processes, which might point towards secondary defects observed in the epididymis (Appendix A). These results support the notion that the induced ROS cascade is not initiated until sperm release from the seminiferous epithelium of the testis.

### 3.5. Blastocysts Stage Embryos Can Be Derived from ICSI of Prm2-Deficient Testicular Sperm

Since *Prm2*-deficient testicular spermatids neither displayed oxidative DNA damage nor severe morphological abnormalities, we tested their fertilization potential *in vitro*. Testicular sperm were extracted from the seminiferous tubules and injected into unfertilized wildtype oocytes using ICSI. Of note, 10% of oocytes injected with *Prm2*-deficient sperm developed into blastocyst stage embryos within four days after injection (Figure 7A,B). Genotyping revealed all embryos to be heterozygous, indicating that *Prm2*-deficient sperm had fertilized the *Prm2*^+/+^ oocyte. (Figure 7C). Control injections with wildtype testicular spermatids displayed similar efficiencies for the induction of blastocyst stage embryos (Figure 7B). Thus, *Prm2*-deficient testicular sperm appear not only morphologically intact but also functionally capable to overcome infertility of *Prm2*^-/-^ males using ART.

## 4. Discussion

In the present study we utilized sperm proteomics to investigate the molecular origin of secondary sperm defects arising in consequence of abnormal sperm protamination. We demonstrate that loss of the sperm`s antioxidant capacity induces a ROS destruction cascade during epididymal sperm maturation, ultimately causing oxidative DNA damage.

The connection of impaired sperm protamination and male subfertility is well-known for more than three decades. However, the molecular processes triggered by this impairment, finally leading to infertility, are mostly unknown, although being essential for the development of targeted treatment options. Here, our *Prm2*-deficient mouse lines serve as an ideal model for functional investigations, as they enable for the first time studies on a genetically and phenotypically uniform sperm population [29]. Protamine-induced DNA hypercondensation is proposed to protect the paternal genome from damaging environmental insults. However, the strong DNA degradation observed in human sperm with abnormal protamination and *Prm2*-deficient murine sperm implies an active destruction cascade as the underlying cause rather than a spontaneously induced damage. It is a longstanding matter of debate, how DNA damage is mechanistically induced [66]. Three mechanisms have been discussed in the past, including apoptotic processes, impaired repair of DNA strand breaks occurring during histone to protamine exchange and harmful effects of excessive ROS level [66]. Here we provide for the first time evidence that *Prm2* deficiency triggers oxidative stress leading to DNA damage, and thus infertility. This was demonstrated by a strong positivity of epididymal sperm cells for 8-OHdG, a well-established marker for an oxidative stress-induced DNA base modification. Longstanding evidence exists for the implication of ROS in male fertility and sperm function. While low levels of ROS are required for sperm maturation, capacitation and acrosome reaction, exceeding levels have been strongly correlated with male infertility and sperm damage [67,68,69]. Besides strong DNA damage, *Prm2*-deficient sperm are completely immotile and display decreased viability due to severe defects in membrane integrity as well as detachment and partial loss of the acrosomal cap. Each of these aberrations is detrimental for physiological sperm function and fertility. These secondary effects are consistent with the known effects of oxidative stress. The sperm membrane contains a large amount of polyunsaturated fatty acids, which are susceptible to ROS-mediated lipid peroxidation, thereby demolishing membrane integrity and perturbing sperm motility in consequence [70]. Further, *Prm2*-deficient sperm display a strongly reduced head size, which has also been reported for sperm from *Prm2* chimeric mice [71]. This is unexpected in light of the DNA condensing function of protamines. However, it is in accordance with a *Gpx5*-deficient mouse model for oxidative stress, showing a decreased sperm volume and surface area as well [72]. Increased ROS levels result either from an increased production or a diminished antioxidant capacity. In *Prm2*-deficient sperm, significantly lower levels of the ROS scavenger enzymes SOD1 and PRDX5 are observed. SOD1 catalyzes the detoxification of superoxide radicals into hydrogen peroxide and oxygen. Studies in knock-out mice revealed that *Sod1*-deficient male mice remain fertile, however, sperm show impaired motility [64,73]. In consequence, in vivo such sperm cannot compete with wildtype sperm and display reduced fertilization ability in IVF experiments [64,74]. A correlation between SOD1 activity and sperm motility as well as DNA fragmentation is also reported for human sperm [75,76]. Alvarez et al. discuss superoxide dismutase (SOD) to be a key player for preventing human sperm from lipid peroxidation [70]. PRDX5 belongs to the class of peroxiredoxins, which act downstream of SOD and catalyze the detoxification of hydrogen peroxide as well as hydroperoxides and peroxynitrites [77]. Six peroxiredoxin isoforms have been identified, with an isoform-specific cellular localization. In humans, PRDX5 is enriched in the postacrosomal region of the sperm head and the mitochondrial sheath [78]. Inhibition of peroxiredoxin activity in murine sperm caused increased ROS levels, thereby underlining its role as antioxidant [63]. In mice, impaired peroxiredoxin activity results in DNA fragmentation and adversely affects sperm motility, viability, fertilization capacity and early embryonic development in mice [63]. In accordance, decreased peroxiredoxin levels are associated with male infertility and oxidative stress-mediated sperm damage in humans [79].

Taken together, the identified molecular changes in the proteome of *Prm2*-deficient sperm nicely delineate the underlying impact of ROS on sperm morphology and function - but how is *Prm2* deficiency sensed and the ROS cascade induced? Interestingly, PRM2 and SOD1 share certain structural similarities. Both proteins harbor a zinc binding-site and are characterized by the formation of intramolecular disulfide bridges. Zinc incorporation into PRM2 is proposed to stabilize the chromatin structure of testicular sperm. However, during epididymal sperm maturation and enhanced disulfide bonding of protamines, zinc is released, as shown in stallion [80]. For SOD1, binding of zinc is essential for enzyme stability and function [81,82]. It is tempting to speculate that zinc ions released from PRM2 might confer stability of SOD1 in the epididymis, thereby ensuring its proper antioxidant function. So, loss of PRM2 might result in decreased zinc concentrations, ultimately causing enhanced degradation of SOD1, leading to an increase of toxic superoxide radicals. The observation that *Prm2*-deficient testes seem to be protected from ROS-mediated sperm destruction in contrast to epididymal compartments requires further analyses. Maybe the total antioxidant capacity [83] of the epididymis is lower, so that the decline of SOD1 and PRDX5 cannot be compensated. Since the epididymis is less immune privileged than the testis [84], the induced ROS cascade might be additionally fueled by infiltrating immune cells.

It remains to be investigated whether antioxidant supplementation would be sufficient to attenuate secondary sperm defects and restore fertility of *Prm2*-deficient males. In fact, antioxidant treatment of human IVF patients has been reported to improve sperm motility and semen parameters [85]. Of note, decreased sperm motility is not solely attributed to *Prm2*-deficient sperm, but has been consistently reported for *Tnp1* and *Tnp2* knockout mice as well as for *Prm1* chimeras [26,86,87]. Although not directly proven, this is likely to be an oxidative stress-mediated response as well. Thus, it is tempting to speculate that a general checkpoint mechanism exists that controls the status of DNA hypercondensation and initiates the ROS response if needed. This response is finally exerted during epididymal transit as verified by the first occurrence of oxidative DNA damage, reduced head size and acrosomal malformations during passage through the epididymis. The preserved integrity of testicular sperm enables new treatment options. Here, ICSI with *Prm2*-deficient testicular sperm initiated proper embryonic development and blastocyst formation. In contrast, ICSI of *Prm2*-deficient epididymal sperm from *Prm2* chimeras mostly resulted in an early embryonic arrest [71]. This confirms that the decline in chromatin integrity during epididymal sperm maturation directly impacts on early embryonic development. Similarly, ICSI with testicular sperm from *Tnp1*^−/−^*Tnp2*^+/−^ males yielded higher implantation rates and more offspring than obtained by ICSI of *Tnp1*^−/−^*Tnp2*^+/−^ cauda epididymal sperm [88]. In men, impaired sperm protamination is associated with recurrent miscarriages [89] and decreased fertilization rates in ART programs [23,24,25]. Thus, in light of the molecular mechanism provided in this study, ICSI with testicular sperm presents as only treatment option for subfertile men with impaired protamination and severe DNA fragmentation. In fact, a recent study proves that testicular sperm from patients with impaired protamination are superior to epididymal sperm in terms of fertilization rate and pregnancy outcome following ICSI [90].

## Figures and Tables

**Figure 1 cells-09-01789-f001:**
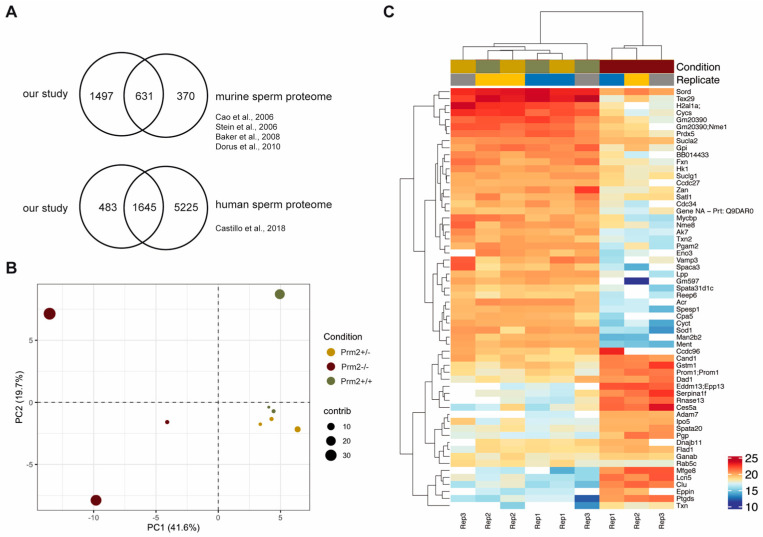
The sperm proteome of *Prm2*-deficient mice. (**A**) Venn diagrams showing the overlap of the generated dataset with published datasets of murine and human sperm. (**B**) Principal component analysis (PCA). MS comprised three biological replicates per genotype. Contribution of each dataset to the PCA is reflected by the dot size. (**C**) Heatmap visualization of the top 50 most significantly deregulated proteins (|log_2_ FC| > 1; FDR < 0.05) between *Prm2*^+/+^, *Prm2*^+/−^ and *Prm2*^−/−^ samples.

**Figure 2 cells-09-01789-f002:**
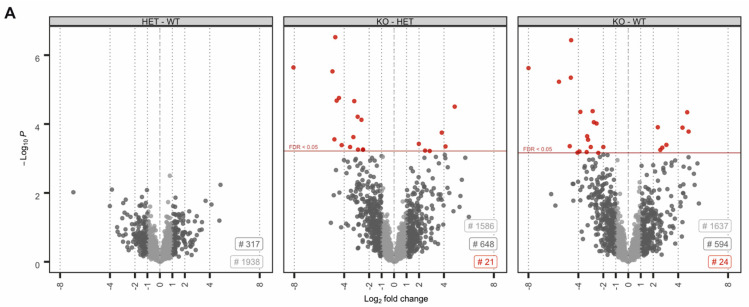
The molecular function of deregulated proteins in *Prm2*-deficient sperm. (**A**) Volcano plots displaying DE proteins in pairwise comparison of *Prm2*^+/+^ (WT), *Prm2*^+/−^ (HET) and *Prm2*^−/−^ (KO) samples. (**B**) Reactome pathway analysis of proteins downregulated in *Prm2*^-/-^ sperm compared to wildtype (|log_2_ FC| > 1; FDR < 0.1). FDR is given in each bar. (**C)** STRING analysis of proteins downregulated in *Prm2*^−/−^ sperm compared to wildtype (|log_2_ FC| > 1; FDR < 0.1). Proteins involved in biological processes of hydrogen peroxide metabolism are highlighted in blue.

**Figure 3 cells-09-01789-f003:**
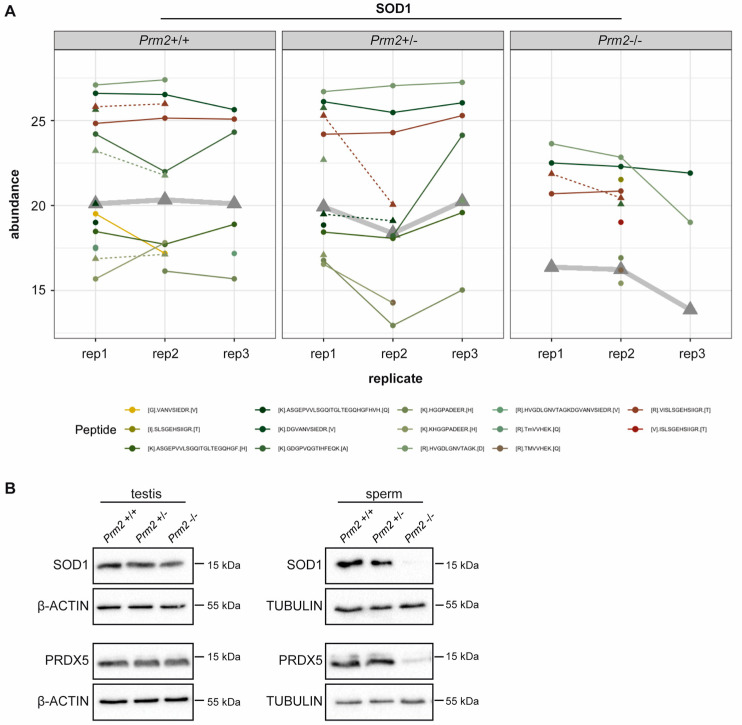
ROS scavenger proteins SOD1 and PRDX5 are downregulated in *Prm2*-deficient sperm. (**A**) Profile plots showing the relative abundance of identified SOD1 peptides within each replicate of *Prm2*^+/+^, *Prm2*^+/−^ and *Prm2*^−/−^ sperm. The protein abundance calculated by Tukey´s median polish is resembled by thickened grey lines. (**B**) Representative immunoblot against SOD1 and PRDX5 on protein lysates from *Prm2*^+/+^, *Prm2*^+/−^ and *Prm2*^−/−^ testes and epididymal sperm (*n* = 1–3). Beta-actin and tubulin served as loading control.

**Figure 4 cells-09-01789-f004:**
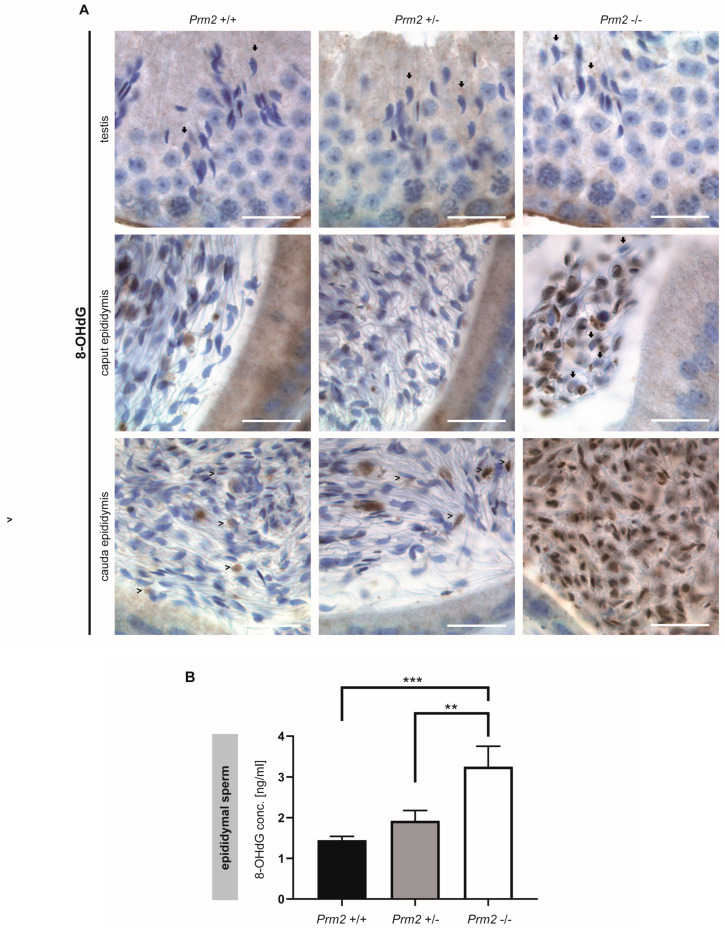
Oxidative stress causes DNA damage in *Prm2*-deficient sperm during epididymal maturation. (**A**) Representative immunohistochemical staining (*n* = 3) against 8-OHdG as a biomarker for oxidative DNA damage. Staining of testicular (top row), caput epididymal (middle row) and cauda epididymal (bottom row) tissue sections from *Prm2*^+/+^ (left column), *Prm2*^+/−^ (middle column) and *Prm2*^-/-^ (right column) animals is shown. Note increasing oxidative damage in *Prm2*-deficient sperm during epididymal transit. Arrows highlight 8-OHdG negative spermatids/sperm, arrowheads 8-OHdG positive cells. Scale bar: 20 µm. (**B**) Quantification of 8-OHdG levels in epididymal sperm DNA from *Prm2*^+/+^, *Prm2*^+/−^ and *Prm2*^−/−^ males by ELISA (n = 5; statistical significance was determined by non-parametric Kruskal-Wallis-Test, ** *p* = 0.0002; *** *p* < 0.0001).

**Figure 5 cells-09-01789-f005:**
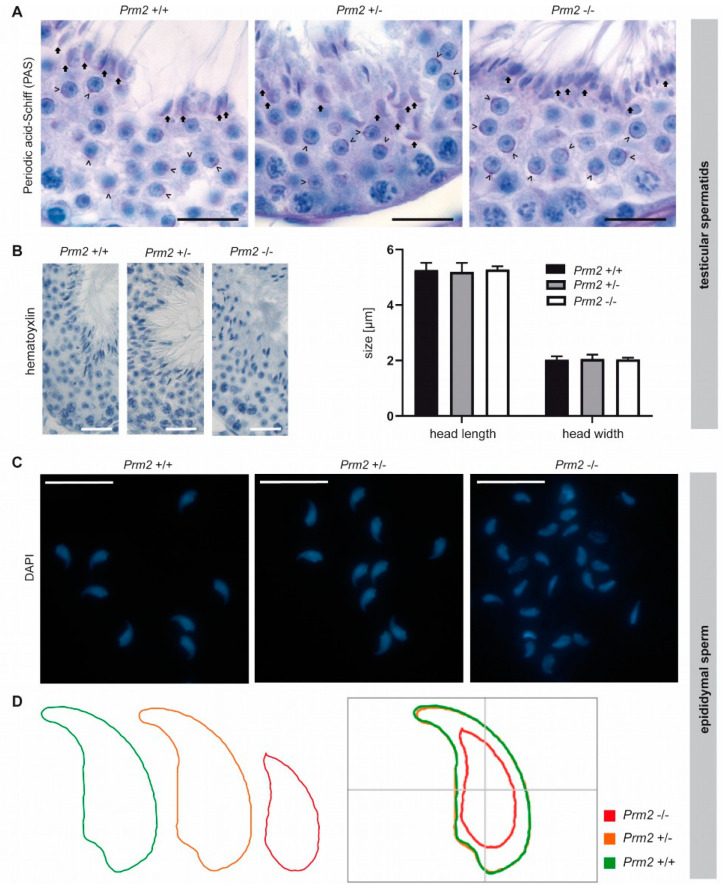
Secondary sperm defects are acquired during epididymal transit. (**A**) Periodic-acid-Schiff (PAS) staining of testicular tissue sections from *Prm2*^+/+^, *Prm2*^+/−^ and *Prm2*^−/−^ males. Representative photomicrographs of stage VI/VII seminiferous tubules are shown. Proacrosomal vesicles of round spermatids are highlighted by arrowheads; mature acrosomes of step 16 spermatids by arrows. Scale bar: 20 μm. (**B**) Representative images of hematoxylin stained testicular step 16 spermatids. Scale bar: 25 µm. Quantification of sperm head length and width. Bars represent mean values ± SD (*n* = 5). (**C**) Representative images of DAPI-stained epididymal sperm from *Prm2*^+/+^, *Prm2*^+/-^ and *Prm2*^-/-^ males. Scale bar: 20 µm. (**D**) Consensus shape computed for *Prm2*^+/+^ (green), *Prm2*^+/−^ (orange) and *Prm2*^−/−^ (red) sperm populations.

**Figure 6 cells-09-01789-f006:**
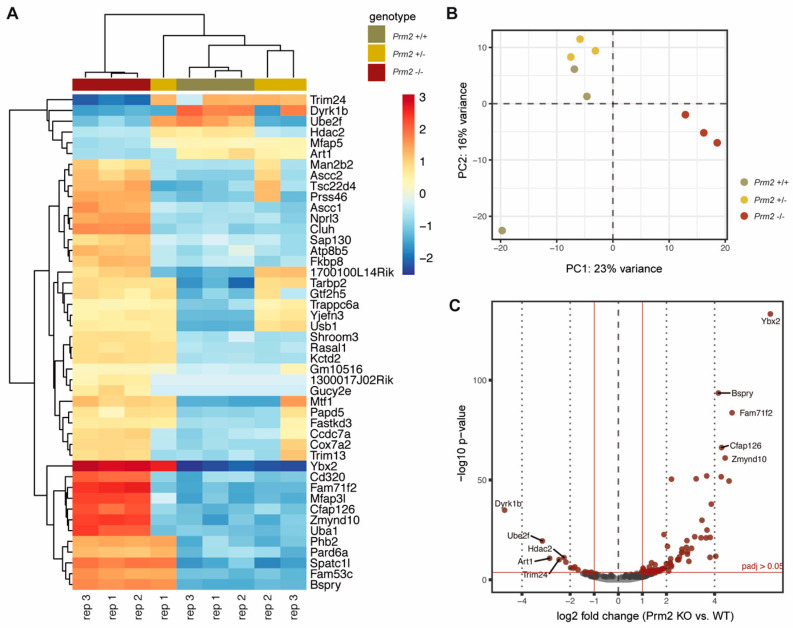
Changes in the testicular gene expression profile of *Prm2*-deficient mice. (**A**) Heatmap visualization of the top 50 DE genes (*Prm2*^−/−^ vs. *Prm2*^+/+^) obtained by RNAseq on *Prm2*^+/+^, *Prm2*^+/−^ and *Prm2*^−/−^ testes. Three biological replicates (rep) were analyzed per genotype. (**B**) Principal component analysis (PCA). (**C**) Volcano plot displaying DE expressed genes (*Prm2*^−/−^ vs. *Prm2*^+/+^) (adjusted p-value <0.05, log_2_ fold change (LFC) >0.5) between *Prm2*^+/+^ and *Prm2*^−/−^ testes.

**Figure 7 cells-09-01789-f007:**
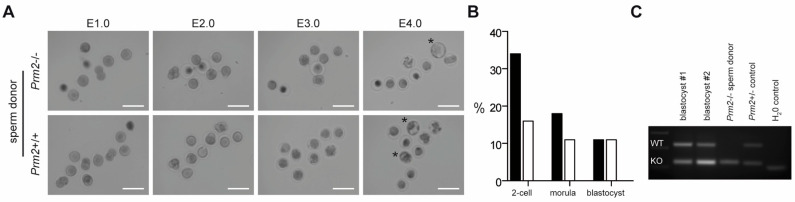
Infertility of *Prm2*-deficient males can be overcome by ICSI of testicular sperm. (**A**) Representative photomicrographs showing the in vitro development of oocytes upon ICSI of wildtype (bottom row) or *Prm2*-deficient sperm (top row) between embryonic day 1 and 4 (E1.0-E4.0). Blastocyst stage embryos are highlighted by asterisks. (**B**) ICSI statistics. (**C**) Genotyping of blastocysts derived from ICSI of *Prm2*^−/−^ sperm into wildtype oocytes. DNA from the *Prm2*^−/−^ sperm donor and a heterozygous animal served as controls.

**Table 1 cells-09-01789-t001:** Antibodies and dilutions used in this study.

Antibody	Company	Product-No.	Dilution (IB)
SOD1	Abcam, Cambridge, UK	ab16831	1:1000
PRDX5	Abcam, Cambridge, UK	ab231892	1:500
α-tubulin	Santa Cruz Biotechnology, Dallas, TX, USA	sc-8035	1:500
β-actin	Sigma, St. Louis, MO, USA	A5441	1:10000
anti-mouse HRP	Dako, Glostrup, Denmark	P0260	1:1000
anti-rabbit HRP	Dako, Glostrup, Denmark	P0448	1:2000

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
