# Peer review of "Protamine-2 Deficiency Initiates a Reactive Oxygen Species (ROS)-Mediated Destruction Cascade during Epididymal Sperm Maturation in Mice"

_cells, 2020, doi:10.3390/cells9081789_

Round 1

Reviewer 1 Report

The manuscript "Prm2 deficiency triggers a Reactive Oxygen Species (ROS)-mediated destruction cascade during epididymal sperm maturation in mice" by Schorle as corresponding author is very interesting.

The aim of the paper is of great interest for the scientific community and it is adequate with the overall Journal scope.

In this study, the mouse model in combination with label-free quantitative proteomics was used to decipher  the molecular origin of secondary sperm defects arising in consequence of abnormal sperm protamination.

The Protamine-2-deficient mouse lines enabled for the first time studies on a genetically and phenotypically uniform sperm population.

The organization and the structure of the article are more than satisfactory and in agreement with the Journal instructions for authors.

The introduction provides sufficient background and the discussion of the results are great.

Reported below few suggestions:

- in the title and in the abstract use Protamine-2 (Pm2) in place of Pm2.

- in the abstract use Superoxide dismutase type 1 (SOD1) and Peroxiredoxin 5 (PRDX5) in place respectively of SOD1 and PRDX5

- in the introduction, lines 69-70 report please only the aim. Delete lines 70-75 from "We show ...

- in the results, line 334 indicate please the sperms in the staining of testicular tissue (top row) using the same darts as in fig 5

-in the references session I suggest to maintain only bibliography from the last 20 years, deleting many dated citations  (see fo example 2, 4, 6, .......) having support to the recent studies done from the journal you have selected or others.

Author Response

Dear Editors of ‘Cells’, dear Reviewers,

thank you for the valuable suggestions, which helped to improve the manuscript. We are re-submitting a highlighted version so that all changes can be traced. In the following you find a point by point discussion of the reviewer comments.

We hope that the manuscript is now suitable for publication and are looking forward to your decision.

Best regards,

Hubert Schorle

please see attached document for detailed point by point reply

Reviewer 2 Report

The influence of impaired protamination on fertility is a well known scientific problem, which is widely commented. In this manuscript Authors try to find the mechanism of ROS-mediated sperm destruction during epididymal sperm maturation in mice with protamine deficiency.  All methods used in these experiments are modern, objective, quantitative and based on statistical analysis data. Authors possess their own excellent model for this study: Prm2-deficient mice lines, generated using CRISP/Cas9-mediated gene-editing. Experiments were conducted on all possible levels from protein analysis to RNAseq  and in the end the functional test ICSI was done. The manuscript seemingly should be of great interest to the readers. Although all experiments were conducted on mice, Authors have introduced the important clinical message for human fertility, that ICSI with testicular sperm may be treatment option for men with impaired protamination and severe DNA fragmentation. However there are many fundamental questions, which should be discussed and explained by Authors because, cause and effect chain is in question.

  1. The active role of protamins has not be demonstrated. Therefore is it better not to use the word „tigger” in title, but indeed „exposure increases” only.
  2. It is not explained, why testis is protected from ROS mediated destruction and epididymis not? Both in epididymis and in testis there is no protection, because in both compartments there are no protamins. On what basis? Maybe „antioxidative buffer” is better in testis than in epididymis? In Discussion section this hypothesis should be widely discussed.
  3. In the proteomic analysis Authors have found that in knockout mice 24 proteins linked with energy metabolism and detoxification of ROS were significantly deregulated compared to wild type. They have chosen SOD1 and PRDX5 which are the most important for ROS detoxification  and revealed during MS analysis that their peptides were strongly diminished in Prm2-/- spermatozo  What about others SOD’s and prominent antioxidants?
  4. The most important is the fact that SOD1 and PRDX5 are downregulated in Prm2-deficient sperm. The validation MS results obtained by Western Blot analysis revealed that SOD1 and PRDX5 were strongly decreased in Prm2-/- epididymal spermatozoa, but not in testicular lysates. The total lysates protein levels were not changed, however there was no comparison: total testicular lysates vs. total epididymis lysates.
  5. In Section 3.4, line 369 „sperm head length and width” was the sperm head or spermatid head measured? What was the proportion of DNA part versus acrosome part? We are concentrated on protamine deficiency and their influence on sperm nucleus morphology is essential. Only the head size was reduced? Maybe heads morphometry is not so important how the proportion of DNA part versus acrosome part? Is it possible to differentiate spermatids from sperm cells? If the whole study is focused on deprotamination and decondesation – the morphology of nucleus should be measured.
  6. Section 3.5 There is no result of ICSI with epididymal sperm. That should be real control for this study.

We should remember that sperm differentiation of spermatozoa is a continuous process, which takes time. Is it not clear and almost not possible to sequestrate testis and epididymis. There are some processes, which are initiated in testis and may be visible freshly in epididymis. Each process is ongoing. Authors did not use fractionated sperm and did not mention differentiated spermatozoa in testis.

Minor comments:

It would have been interesting to know, how many mice were included in every step of this study? What was the performance of the protein isolation from mouse sperm?

Authors have reported that Prm2+/- males were fertile  and spermatozoa did not differed morphologically and functionally from wild type spermatozoa. Are the concentration of epididymal spermatozoa in Prm2-/- mice different from wild type mice? Authors have written that „loss of PRM2 did not affect the efficiency of spermatogenesis”. Was the concentration of retrieved epididymal spermatozoa calculated?

The method of elimination of other cells from epididymis is crucial. Are the authors sure that method of  hypotonic shock to induce lysis of non-sperm cells is sufficient?

Author Response

(The authors gave the same response as above.)

Round 2

Reviewer 2 Report

Thank you for discussing the most important questions and taking my advice into consideration.